# Recent Advances in DNA Nanomaterials

**DOI:** 10.3390/nano13172449

**Published:** 2023-08-29

**Authors:** Incherah Bekkouche, Maria N. Kuznetsova, Dovlet T. Rejepov, Alexandre A. Vetcher, Alexander Y. Shishonin

**Affiliations:** 1Nanotechnology Scientific and Educational Center, Institute of Biochemical Technology and Nanotechnology, Peoples’ Friendship University of Russia n.a. P. Lumumba (RUDN), Miklukho-Maklaya St. 6, Moscow 117198, Russia; kuznetsova_mn@pfur.ru (M.N.K.); redzhepov-d@rudn.ru (D.T.R.); 2Complementary and Integrative Health Clinic of Dr. Shishonin, 5, Yasnogorskaya Str., Moscow 117588, Russia; ashishonin@yahoo.com

**Keywords:** nanomaterials, DNA, aptamers, microcircles, nanomaterials, plasmid

## Abstract

Applications of DNA-containing nanomaterials (DNA-NMs) in science and technology are currently attracting increasing attention in the fields of medicine, environment, engineering, etc. Such objects have become important for various branches of science and industries due to their outstanding characteristics such as small size, high controllability, clustering actions, and strong permeability. For these reasons, DNA-NMs deserve a review with respect to their recent advancements. On the other hand, precise cluster control, targeted drug distribution in vivo, and cellular micro-nano operation remain as problems. This review summarizes the recent progress in DNA-NMs and their crossover and integration into multiple disciplines (including in vivo/in vitro, microcircles excisions, and plasmid oligomers). We hope that this review will motivate relevant practitioners to generate new research perspectives and boost the advancement of nanomanipulation.

## 1. Introduction

The multidisciplinary scientific topic of DNA-NMs applications gained popularity in the last century when the use of DNA as a building material was shown to be both brilliant and straightforward. Nearly 40 years of productive effort have solidified the underlying theories of the field and have widened previously unimaginable paths such that now, DNA objects of nearly any desired size and form can be quickly created. More importantly, they can be created with spatial control at the nanoscale level [1]. These efforts resulted in the creation of a macroscopic 3D crystal that was solely built from the self-assembly of a rationally designed DNA tile [2], which proved that it is possible to bridge the gap between the nano- and macroscopic worlds in a programmed manner. Reconfigurable modules were added to various parts of the DNA nanostructure, which was a significant advancement [3]. UV irradiation [4], pH [5,6], and other factors all influence the selective switching between two states of DNA [7,8,9], and these variables are employed to coordinate and frequently reversibly start the global mechanical alteration of the structure. Thus, control over the matter distribution may be used at various periods, which, as stated by Seeman, allows one to “place whatever molecule you want, where you want, when you want it there” [10].

Over the last ten years, a variety of biological applications have made extensive use of the manipulation and characterization of biological cells based on this technique, including cell transfer, isolation, immobilization, and injection [11]. Micro/nano features can be seen in biological substances, cells, and life processes. At the micro- and nanoscale, artificial technologies may communicate with living things in proximity; they can offer fresh scientific understandings and create brand-new instruments for identifying and treating diseases [12].

## 2. Goals of DNA-NMs Development

DNA-NMs have been created and customized to achieve a variety of biomedical applications because of their inherent dynamics and biocompatibility. Several DNA nanodevices have so far been successfully created for performing a wide range of jobs, including sensing tasks and information processing in a biological context, the quantitative imaging of analytes in live cells [13], and disease diagnosis [14].

### 2.1. Biomedical Goals

As indicated in Figure 1 (e.g., [15] modified by authors), biomedical nanotechnology is devoted to investigating nanoscience and nanotechnology for health well-being with the ultimate objective of providing individualized health management. The major role that technology plays in illness monitoring, treatment, and the control of progression has been validated by reports that have been made public by health organizations. Additionally, the application of nanotechnology-assisted methods improves the sensitivity, affordability, and accessibility of illness diagnosis and treatment methods [16,17,18].

The programmable performance of nanosystems, which are researched for biomedical applications, is useful for the design and development of medicines that take patient profiles into account, i.e., individualized health management [19,20], in addition to the development of key characteristics of methodologies that are helped by nanotechnology, numerical approaches, or artificial intelligence [21,22]. Combining bioinformatics, deep learning, and machine learning [23] has become a very effective tool for figuring out patterns and predictions; the understanding of pandemic variance, the improvement of medication, and risk assessment are all aided by this knowledge. A fast and successful analysis, however, depends on the proper administration of bioinformatics [24,25]. A technique in biomedical nanotechnology for data exchange, storage, and analysis called the Internet of Medical Things (IoMT) is now receiving a large amount of attention [26,27,28], particularly for opportunities that are associated with the employment of artificial intelligence (AI).

### 2.2. Nanotechnological Goals

Various scientific research fields for assembling and synthesizing materials, such as synthetic chemistry [29], supramolecular assembly [30], materials processing [31], and bio-related technology, have undergone continuous developments and seen successful advances in tandem with the quick advancements made in the field of nanotechnology. It is possible to contribute to the architecting of materials from nanoscale units by using the scientific concepts of these accomplishments. A creative idea should be put out to merge these study disciplines with nanotechnology for this new age to match the remarkable impacts witnessed by the proposal of nanotechnology (the 21st century) [32]. This was the idea behind nanoarchitectonics, which was first put out in 2000 at the first international symposium on nanoarchitectonics using suprainteractions in Tsukuba, Japan by Masakazu Aono beyond nanotechnology, and this idea has the potential for various fields of science and technology [33].

The nanoarchitectonics concept is supposed to involve the architecting of functional materials using nanoscale units based on the principles of nanotechnology. The production of functional materials via nanoarchitectonics approaches is done through the combination and selection of several unit processes, including molecular synthesis, atom/molecular manipulation, self-assembly/self-organization, field-applied assembly, micro-fabrication, and bio-related processes [34]. Since these procedures may be used for a variety of targets, from physical devices to biological purposes, independent of the material or purpose, they should be applicable to a wide range of targets [35]. The terms “nanotechnology” and “nanoarchitecture” refer to broad, all-encompassing notions. As a result, there is a lot of overlap and a lot of problems that they have in common, both nanotechnology and nanoarchitectonics can be held responsible for the creation of some functional nanostructures and their functional outcomes; their fundamental characteristics and definitions, however, differ: nanotechnology is the handling of nanoscale items, whereas nanoarchitectonics is a process for creating materials and systems out of nanoscale objects [36].

## 3. Technology of DNA-NMs

### 3.1. In Vitro Applications

The need for fresh views on in vivo-like and animal-free methodologies for chemical and pharmaceutical safety evaluation is expanding both in drug development and chemical risk assessment [37].

#### 3.1.1. Enzymes and Their Application In Vitro

Enzymatic catalysis has used scaffolds as reactors. Potential reaction cascades are made possible by these materials’ facilitation of the regulation and improvement of catalyzed reactions under the control of the nanoparticle, enzyme nanoreactors allow researchers to control how much an enzyme can catalyze, as opposed to traditional homogeneous experiments in solution. This method improves the stability and activity of enzymes, enabling the use of enzymatic processes in novel environments [38]. Additionally, numerous enzymes can be compartmentalized in enzyme nanoreactors to regulate biological cascades [39].

The first immobile Holliday junction, which consists of four different strands entwined at a single crossing point, was designed and published by Nadrian Seeman in 1982 [40]. A few years later, the same concept was used to the realization of an indefinitely large periodic network by combining complementary sticky ends that protruded from the tile’s opposing arms. The concept of using DNA as a building material proved to be both clever and straightforward, heralding the birth of a new scientific discipline currently referred to as structural DNA nanotechnology [41]. Currently, DNA objects can be created quickly and, most critically, with precise spatial control at the nanoscale scale level, they can also be created in practically any size and shape [1].

Several years after the foundation of DNA nanotechnology, a different self-assembling technique was created and eventually dubbed scaffolded DNA origami [14]. The fundamental difference between the two assembly procedures hinges on the presence or absence of a lengthy single-stranded DNA (termed scaffold or template) in the reaction mixture (Figure 2) (e.g., [42]) (modified by authors). The relatively small oligonucleotides (20–40 bases long) used in the Seeman original tile-based method are designed to entangle one another into branching motifs of certain geometry [43], according to the somewhat rigorous notion of sequence symmetry minimization [44]. This concept states that sequences involved in the construction of the motif are chosen to be as dissimilar from one another as possible, hence, minimizing the competitive production of alternative secondary structures (Figure 2a). The very tiny size and considerable structural flexibility of such tiles unavoidably lead to their partial dissociation at room temperature, despite thorough design efforts, tiles can be joined into enormous periodic lattices using the cohesion of their complementary adhesive ends to get over this issue. As an alternative, it is desirable to harness the inherent dynamics of such tiny DNA motifs to create DNA circuits and complicated reaction networks, whose kinetic behavior is controlled by foreseeable single-strand displacement processes [3,45,46,47].

DNA-NMs that involve proteins (especially enzymes): from the synthesis of DNA-tagged proteins (herein referred to as DNA–protein conjugates) until their integration onto substantial DNA scaffolds (resulting in what is herein referred to as DNA–protein nanostructures or NMs), with a focus on their binding affinity and catalytic activity. The development of a homogeneous chemical species with a high purity grade is the first essential step in the study of such hybrid materials. Only then can any observable change in the biophysical characteristics of the DNA–protein (enzyme) nanostructure be linked to a specific structural characteristic of the sample [42].

So far, several strategies have been created and are being continuously improved to allow the binding of a protein of interest to a DNA molecule. For creating DNA–protein hybrids, either natural proteins or their genetically modified variations are employed as starting points for chemical conjugation, enzymatic ligation, or a combination of the two. The power and selectivity of the newly created interaction serve as a simple means of categorizing these tactics. It has been proposed that the option of the DNA–protein (enzyme) contact be either covalent, which would be primarily strong and irreversible, or non-covalent, which would be weaker and more readily dissociable. Moreover, the newly created bond may address several sites that are chemically equivalent and physically indistinguishable, or it may only address one specific site of the proteins of interest (POI) [42,48].

The recent finding that single-stranded DNA (ssDNA) and CNTs may be assembled in a sequence-dependent manner has spurred the development of methods for generating long strands of ssDNA (>100 nt) with high yield and homogeneity. Small repetitive DNA sequences have drawn more attention due to their potential applications in nanotechnology as well as the fact that the presence of these repeats often leads to genomic instability in organisms [49]. Twelve disorders were involved by repetitive DNA sequences, such as Friedreich’s ataxia, Huntington’s disease, and myotonic dystrophy, which are caused by repetitive DNA sequences [50,51,52,53,54]. According to recent theories [55,56,57], the creation of non-B-form secondary structures in the repetitive region is the molecular cause of genomic instability. Current theories postulate that the creation of non-B-form secondary structures in the repetitive region is the molecular cause of genome instability; comparable structures also work against the direct chemical production of lengthy repetitive oligonucleotides. Short single-stranded repeated sequences have only been used as model systems for in vitro investigations thus far due to the technical difficulties of generating large, repetitive ssDNAs by normal chemical methods for the creation of ssDNA from double-stranded (ds) products; thus, many techniques have been developed. One of these techniques is affinity purification [55,56,57,58], polymerase chain reaction (PCR) using chemically modified primers that result in the synthesis of strands of uneven length [59,60,61], and preferential exonuclease digestion of one strand [62,63]. Although these procedures are straightforward in theory, exonuclease digestion results in non-uniform populations of ssDNA, while the other procedures are costly, involve several manipulation steps, and yield small amounts of ssDNA. It offers a quick and effective modified asymmetric PCR approach that makes it possible to synthesize ssDNA sequences that are far longer than those made possible by chemical oligonucleotide synthesis techniques. The strategy that has been proposed is based on a three-step process that creates lengthy ssDNA with both repetitive and nonrepetitive sequences in a reliable manner. Initially, under highly optimized circumstances, double-stranded template DNA is produced by PCR. Under the same circumstances, a second PCR reaction utilizing a single primer and a small portion of the previously created double-stranded PCR result produces single-stranded DNA [64,65,66].

#### 3.1.2. Chemical Application In Vitro

NMs are distinctive from bulk materials because of their adjustable and distinctive physical, chemical, and biological characteristics. Chemical and nanotechnology might be researched in a cooperative, synergistic way due to their comparable aims and toolkits. Artificial nanomaterials can affect biological functions on living cells’ surfaces, inside them, and even in certain intracellular compartments. A wide range of exploratory areas, including chemical-, cell-, and mechanobiology, have been made possible by size, physicochemical control, and ensuing biological interactions with nanomaterials. Inorganic to biodegradable polymers, among other wide groups of nanomaterials, have been created for molecular transport, enzymatic catalysis, molecular imaging, and molecular sensing. Numerous uses of nanomaterials have also entered the clinic [67].

DNA has developed into a new, multi-disciplinary area that combines elements of chemistry, biology, biomedicine, and materials science [68]. Compared to synthetically obtained nanomaterials such as carbon nanotubes, quantum dots, and gold nanoparticles [69,70,71,72,73], the biocompatibility, programmability, and addressability of DNA nanostructures are all highly regarded. These characteristics make them potential candidates for cutting-edge biological and biomedical applications [68,74].

##### Aptamers

Short oligonucleotides known as aptamers are selected from a large nucleic acid library using the SELEX method based on their affinity to target cargos [75,76,77,78,79,80]. These target cargos can be incredibly varied, ranging from cells, viruses, and tissues to tiny molecules, amino acids, peptides, and proteins [77,81,82,83,84,85,86]. Aptamers can attach to their targets by noncovalent interactions such as hydrogen bonding, hydrophobic effects, and van der Waals force [86,87], by folding into certain secondary/tertiary conformations. As the method of molecular recognition is comparable to interactions between antibodies and antigens, aptamers are also referred to as “chemical antibodies” [77]. Aptamers have comparable binding affinities and specificities. They can be automatically and repeatedly synthesized on a DNA/RNA synthesizer, can easily and specifically be modified with different functional groups at specific sites, and are typically 5 to 10 times smaller than antibodies. These characteristics may result in low immunogenicity and quick tissue penetration and make aptamers perfect molecular instruments for biological and biomedical applications [77,86,88,89,90].

Cell-SELEX, a cutting-edge method created in 2006 by Tan’s team, enables the selection of aptamers against target living cells without the need to be aware of the chemical fingerprints on the cellular membrane [85,91]. Based on this technology, hundreds of aptamers have been chosen against different cell lines, particularly cancer cells, such as XQ-2d for pancreatic ductal adenocarcinoma and DML-7 for human prostate cancer (DU145) [91]. Other examples include sgc8 and sgc3 for the human acute lymphoblastic leukemia cell line (CCRF-CEM). The number of aptamers and their targets will continue to grow because of cell-quick SELEX’s development. Several engineering techniques have also been developed to improve the affinity and specificity of aptamers or to provide them with custom functionalities [92]. Because of these distinguishing characteristics, aptamers have demonstrated a wide range of applications in biology and biomedicine [85,91,93].

##### Integration of Aptamers in DNA-NMs

The majority of techniques for creating aptamer-integrated DNA-NMs are based on nucleic acid hybridization. Since DNA-NMs are highly programmable and adhere to the rigorous Watson–Crick pairing requirements, aptamers may be added site specifically to provide customized capabilities. This can be accomplished either by post-functionalization or the DNA nanostructure construction technique. For instance, aptamer-modified staples were one of the building pieces used by Tanner and colleagues [94] to construct a DNA nanobox. By combining only the scaffold and staple DNAs, one may create an aptamer-integrated DNA origami. Yang and colleagues [95] constructed a nanostructure with a sticky end, then hybridized it with the extended strand of the aptamer to create an aptamer-guided DNA tetrahedron. A wide range of aptamer types and quantities may be used to effectively insert aptamers into the surface or wireframe of DNA nanostructures. For instance, Willner and colleagues [96] created a diverse sensing platform by integrating several aptamers with a DNA tetrahedron. Chu and colleagues [97] placed 12 aptamers on 2 sides of sheet-shaped DNA origami for targeted drug administration. A few non-hybridization approaches have also been described for the building of aptamer-integrated DNA-NMs, in addition to hybridization assembly, as, although creating such objects is not too difficult, ensuring their stability is still a problem. Their integrity would be impacted by a variety of variables, including temperature, pH, and metal ion concentration. Moreover, nuclease degradation frequently poses a severe issue, particularly in biological environments [98]. Many engineering techniques have been developed to increase the stability of aptamers-integrated DNA-NMs. For instance, a circular bivalent aptamer was created by joining the 3′-OH and 5′-phosphate groups of two aptamers (Figure 3a), which demonstrated increased enzymatic resistance to Exo I [99]. In a study published by Sczepanski and colleagues [100], the nuclease-resistant aptasensor that could maintain full functioning in the blood for 12 h was designed using peptide nucleic acids (PNAs) and l-ribose-based nucleic acids (l-RNA). More modified nucleotides and synthetic bases have been generated with the advancement of bioorganic chemistry, adding to the genetic information and generating some new biological functions [101,102,103,104,105,106]. For instance, Wang et al. [104] created and produced some 3,5-bis(trifluoromethyl)benzene-containing fluorinated nucleic acids. These synthetic bases improved the thermostability and resistance to cell lysis of molecular beacon probes (Figure 3b) [104]. By combining azobenzene with a natural T base, the same group also created a photoresponsive base (zT) (Figure 3c) [107]. It might be utilized as a brand-new molecular or functional “element” to create innovative biomaterials with distinct qualities. More recently, putting thymidines close together and subjecting them to UV light produced a covalent cyclobutane pyrimidine dimer [108]. With these extra connections, DNA origami showed improved resistance to nuclease and could withstand temperatures of up to 90 °C in pure water. Moreover, Shih and colleagues [109] discovered that the PEGylated oligolysine-coated DNA origami could survive for more than 48 h when incubated with 2600 times the normal dose of DNase to crosslink these nanostructures, glutaraldehyde was utilized (Figure 3 (e.g., [109,110]) (modified by authors)).

### 3.2. In Vivo Applications

DNA-NMs are appealing drug delivery vehicles due to their biocompatibility, biodegradability, and nontoxicity [111,112], and the capacity to create customizable 2D and 3D structures. Furthermore, DNA strands are easily altered chemically and physiologically, making them suitable for use as drug delivery systems [112,113], artificial lipid membrane channels [114], and platforms for enzymatic and chemical reactions [115], DNs are far more addressable than traditional carriers, which gives them the following benefits: They are monodisperse, which enables the precise spatial arrangement of a variety of ligands with controlled valency, and they can be modified with functional moieties on their surface to affect their biological function. They also allow control over their size and shape to suit different cargoes (even within the same carrier).

The dense packing of DNA helices required for the self-assembly of DNA-NMs frequently causes electrostatic repulsion between the negatively charged phosphate backbones of the DNA helices. High concentrations (5–20 mM) of divalent cations, such as Mg^2+^, are necessary for structural stability to prevent DN denaturation. When compared to physiological fluids like human serum, which typically contains 1 mM of divalent cations, these concentrations are about an order of magnitude greater [116]. Furthermore, DNs are especially vulnerable to nuclease action and risk destruction in cell medium and in vivo. Numerous investigations have demonstrated that when DN is incubated with 10% fetal bovine serum, nuclease activity causes fast breakdown [115,117].

### 3.3. In Situ Applications

It is crucial to monitor and control chemicals in organisms in real time at the molecular level. Because of this, in situ target biomolecule analysis has enormous promise for physiology and medical research. Through DNA sequence coding, the decades-old DNA nanotechnology developed by Seeman opens new opportunities for the synthesis and creation of synthetic DNA-NMs in vitro [40]. DNA nanotechnology has made significant advancements in biosensors, biological detection, and drug administration thanks to its inherent biocompatibility and high membrane permeability [118,119]. Particularly, DNA-NMs as probes provide an effective tool for in situ bioanalysis by making use of remarkable molecular recognition characteristics [120,121], nucleic acid probes in vivo are easily susceptible to many aspects including probe integrity, activity, and conformation. Additionally, the target accessibility reduction of biosensors is also a dominant obstacle. In an effort to develop DNA nanotechnology, using structural DNA or DNA-NMs as probes signally improves the stability of the probes and provides a confined nano-space to increase the specificity and sensitivity of the analyses [122].

### 3.4. Other Examples of Self-Assembled DNA-NMs

In addition to structures obtained in Seeman’s lab, there are some other recently developed structures worth mentioning, in particular, if employing polycations to assist the shaping.

By complexation of plasmid DNA by PEG-polycationic block copolymers, its self-assembly results in either toroidal structures, rod structures, or globular structures. The outcome depends on salt concentration and strandedness of DNA, e.g., the complexation of dsDNA with PEG-polycation in 600 mM NaCl forms toroidal structures. Without salt, the same dsDNA forms rod-shaped structures. Compared to the rod-like structure, the toroidal structure has better biological functions that can not only increase transcription in vitro but also increase the efficiency of gene transduction in vivo. This demonstrated the great usefulness of toroidal packing of plasmid DNA as a separate structured carrier of genes [123]. The explanation of differences is based on the peculiarities of the interactions between DNA and the copolymers, and the specific nature of dsDNA in rigidity, nucleation, and growth for polyplexes of low molecular weight polycations [124]. Short ssDNA forms with short polycations just spherical structures (micelles) [125,126]. However, it should be mentioned that the elongation of ssDNA inevitably leads to complex secondary and ternary structure formation and structurization of its assemblies with polycations [126].

The simplicity of manipulation of polymer micelles makes it possible to adjust important parameters necessary for the protection of NA and effective delivery to the target sites. The ordered core–shell structure allows the inclusion of other therapeutic/diagnostic agents, providing an opportunity for combined therapy and theranostics [127].

By self-assembly of DNA structures, most DNA hydrogels are formed. Self-assembly can occur in a polymer matrix to build an interpenetrating network (IPN) hydrogel. The study of the mechanism shows that self-assembly between short chains of ssDNA leads to the formation of long linear DNA parts and loop structures, and entanglement between these DNA assemblies at high concentrations leads to the formation of three-dimensional hydrogel networks. By programming the sequence of the ssDNA chain, it is possible to construct a pH-sensitive DNA hydrogel or a catalytic DNA hydrogel [128]. Programmed self-assembly of nucleic acid strands proved to be highly effective for creating a wide range of structures of the desired shape [129].

Polyhedral DNA-NMs consisting of three- or four-branched branches connected by rigid ribs in the form of a double helix can be constructed using self-assembly protocols that use the specificity of base pairing interactions. The DNA tetrahedron was the first such structure to be created quickly and with high yield from multiple chains in a single assembly step. The bipyramid contains three four-shoulder and two three-shoulder joints; all its faces are triangles, so the design must be rigid. Easy and efficient synthesis is a necessary step toward using these controlled three-dimensional frameworks as building blocks for functional molecular cell construction [130].

### 3.5. DNA Adsorption over the Nanoparticles

As mentioned above, DNA-NMs are of great interest mostly because such hybrid materials combine the molecular recognition and programmability of DNA with the physical properties of organic or inorganic nanoparticles, demonstrating promising applications in many fields, including biosensing, drug delivery, materials science, and nanotechnology. Over the past two decades, many nanomaterials such as metal nanoparticles, semiconductor quantum dots, and nanoscale carbon materials such as carbon nanotubes (CNT) and graphene oxide have been modified by DNA.

The key step in creating such materials is the attachment of DNA to the surface of the particles. While covalent attachment provides a strong bond with DNA, physical sorption is attractive for its simplicity, cost-effectiveness, and reversible binding [131].

Understanding the processes of adsorption and desorption is necessary for the optimization of DNA microarrays, the development of biosensors, and the functionalization of nanoparticles. Therefore, the interaction of DNA with inorganic surfaces aroused great research interest [121]. Adsorption of ssDNA on gold nanoparticles proceeds more efficiently than adsorption of dsDNA. ssDNA forms multipoint contacts with AuNP, and the nucleotide sequence, its length, the presence of modifications in DNA, the ionic strength and pH of the medium during incubation, the temperature, and reaction time of the association significantly affect the efficiency of the observed adsorption of cDNA to the surface of AuNP. It is shown that the adsorption of various DNA structures (dsDNA, ssDNA with a hairpin structure, and self-complementary ssDNA) on AuNP really differs significantly from nanoparticles [132].

There are two main differences between ssDNA and dsDNA in the context of their interaction with AuNP. Firstly, heterocyclic bases of ssDNA, being spatially accessible, are capable of forming hydrogen bonds and/or hydrophobic contacts with atoms on the surface of the AuNP; these processes are very difficult for heterocycles inside dsDNA. According to a number of studies, it is this interaction that determines the binding strength of NA with AuNP. The second difference is that ssDNA has a much more mobile structure, whereas dsDNA has a rigid structure that prevents direct interaction of nitrogenous bases with the surface of nanoparticles [132].

ssDNA can be adsorbed by AuNP. Coating nanoparticles with citrate leads to an increase in the stability of AuNP. Studies have shown that the adsorption capacity of DNA is determined by the length of the DNA oligomer and the concentration of DNA and salt since both DNA and AuNP have the same charge in aqueous solutions [133]. DNA adsorption on AuNP, therefore, could also be achieved by the use of buffers with low pH and high alcohol concentrations [133] and freezing [134].

AuNPs, functionalized with DNA, are widely used for targeted assembly of materials, biosensors, and drug delivery. All four DNA bases can be strongly adsorbed on gold due to coordination interactions, and therefore even non-isolated DNA can be attached to AuNP [135].

## 4. Products of DNA Nano-Tuning

### 4.1. Plasmids

#### 4.1.1. Plasmid Oligomers

Nucleic acids must self-assemble to support many of the fundamental biological activities in living things. The storing and programmed execution of instructions stored in a DNA sequence are key aspects of this self-assembly process [136]. It is interesting to note that just four nucleobases—A, G, C, and T/U—interact non-covalently in these instructions. Remarkably, this small collection of recognition units has acted as genetic material during billions of years of evolution. Yet, new kinds of building blocks could lead to DNA analogs with traits that diverge significantly from the natural archetype. The idea has served as the inspiration for various attempts to add structural variation to the nucleotide, the smallest constituent part, to increase the functioning of nucleic acid-based, self-assembled systems [137].

#### 4.1.2. Plasmid Microcircles Excision

Microcircles, a prokaryotic vector component made up of one active transcriptional unit plus an attachment region to the matrix (S/MAR), are described as the result of parental construct optimization. Some changes still exist, but this name is still used to refer to vectors with a comparable function [138] (Figure 4). Such vectors, which are topologically closed circular DNA, have some biopharmaceutical applications [139]. A transcriptional unit or a region of attachment to the matrix cannot be contained inside such circles, it is important to note. Most surprisingly, although they are just a little bit longer than 80 bp, they are still referred to as little circles. They recommended referring to them as microcircles, nevertheless, as they are two orders of magnitude smaller than the minicircles [140]. In the future, circular DNA shorter than 1 Kbp in length will be referred to by this name. Nowadays, the name “microcircles” refers to topologically closed DNA created from synthetic oligos using specifically created methods [141], which enable the synthesis of microcircles between 42 and 75 bp. The need for a new definition of microcircles is made even more pressing by the fact that some writers use it to refer to circular recombination products with 400–600 bp but put it in quotation marks for no apparent reason [142].

### 4.2. DNA-CNT Hybrids

When Zheng et al. presented the hybridization of single-stranded DNA (ssDNA) and single-walled CNTs (SWNTs) in 2003, it was the first time that DNA and CNTs have been reported to interact [143]. In addition to presenting actual results, they also offered a theoretical model of DNA adsorption on SWNT surfaces (Figure 5) (e.g., [143,144]) (modified by authors). Additionally, it was hypothesized in this study that there is a connection between SWNT chirality and certain bases in DNA molecules. Use of double-stranded DNA (dsDNA) in 2003 to independently characterize the solubility of SWNTs with DNA molecules. Following that, many research teams have reported on the hybridization of DNA with CNTs, particularly SWNTs.

Since 2003, two primary hybridization research objectives have been established. Enhancing CNTs’ solubility is the first step. Uncoated CNTs often form insoluble bundles in aqueous solutions, even though their distinctive structural and physicochemical features make them intriguing for both basic research and commercial applications. CNTs need to be produced in a mono-dispersed state to maximize their potential, and this is particularly true for SWNTs. Mono-dispersed CNTs can be made by attaching hydrophilic organic molecules, which can improve CNT solubility. Examples of these molecules include sodium dodecyl sulfate (SDS), peptides, proteins, and DNA molecules.

When biomaterials (such as proteins, enzymes, antigens, or DNA) are combined with CNTs, new hybrid systems are created that combine the conductive or semiconductive capabilities of CNTs with the catalytic or recognition abilities of the biomaterials. New bioelectronic systems, such as biosensors and field-effect transistors, or nanocircuitry that is templated, may result from this. The development of this research area involves a variety of difficult problems, including the synthesis of site-specific and structurally defined biomaterial–CNT hybrids, enhanced techniques for the separation and characterization of conductive and semiconductive CNTs, the ordered and controlled assembly of addressable biomaterial–CNT systems on surfaces, and the creation of microscopic imaging methods to characterize the structures and functions of the nanoscale devices [145].

### 4.3. DNA Origami “Capsidisation”

The DNA origami technique makes it possible to form arbitrary, precise, and complex two- and three-dimensional nano-objects with specified rotations, curvature, and tension. Traditionally, origami is formed by folding a skeleton thread, that is, a long ssDNA, into the desired shape using a predefined set of oligonucleotides, but constructions without frameworks are also possible [146].

Employment of DNA origami nanostructures as platforms for binding and assembly, it is possible to achieve control over the shape, size, and topology of the viral capsid. The latter can have different geometries, most clearly characterized by icosahedral or spiral symmetry. It is important to note that precise control over the size and shape of viral capsids will be important when developing new vaccines and delivery systems. In addition, the resulting viral capsid coatings can protect encapsulated DNA origami from degradation [147].

The transfection efficiency could be gradually increased with increasing concentration to a level at which the DNA origami was completely covered with capsid proteins. Complete encapsulation makes it possible to deliver multiple functionalized DNA origami to cells and, thus, use programmable combinations of specific drugs for achievable medical procedures. In the future, the organization of reactions inside the cell can be realized with the help of modular docking nodes on origami [146].

### 4.4. Spherical NA

Finally, it is worth mentioning spherical nucleic acids, which are structures composed of chemically modified inorganic nanoparticles in the nucleus and a dense layer of highly organized thiol-modified oligonucleotides as a shell, which are chemically bound to the surface of the nucleus through thiol bonds. Oligonucleotides provide a negative charge and increased colloidal stability, steric stabilization in solutions with increased ionic strength, and the ability to combine with other biomolecules for molecular visualization or drug delivery [148].

The developed antiviral SNA vaccine with the employment of spherical NA induces the formation of memory B cells in human cells. Its administration to mice in vivo creates stable binding and neutralizing antibody titers, mice that received the SNA vaccine had a 100% survival rate and lungs that were free of the virus by plaque analysis [149].

## 5. Conclusions

DNA is one of the most advantageous nanotechnology materials. It can self-assemble, build programmable nanostructures, act as a platform for mechanical, chemical, and physical devices, and have the ability to self-assemble. Even though nature has evolved many intricate nanoscale systems over thousands of years, scientists and engineers must work hard to create new technologies that will enable the use of DNA in more fields, including physics, medicine, computing, and material sciences. Nanotechnology must be enhanced to fulfill the need for better detectors in the sectors of biological and chemical detection as well as for higher sensitivity. More efforts are necessary for the further development of ligands functionalized DNA NM, even if nano–DNA manipulation has demonstrated potential qualities for targeted treatment and diagnosis:The stability of DNA nanostructures should be improved with great care. It is also necessary to develop techniques for examining the stability of DNA nanostructures in vivo. It is known that unprotected DNA experiences degradation in vivo [150].It is necessary to make it easier to manipulate ligands on DNA nanostructures, increasing the efficiency of DNA–ligand conjugation and identifying the distribution of receptors on the cell surface. The creation of ligand-conjugated DNA nanostructures requires the development of effective preparation methods.It is important to assess the in vivo pharmacodynamics and pharmacokinetics of ligand-functionalized DNA nanostructures. The application of therapeutic ligand modification on DNA nanostructures is common. Nonetheless, in vitro research is the main emphasis of the effort.

One of the most promising platforms for the modification of ligands to support the advancement of precision medicine is DNA nanotechnology.

## Figures and Tables

**Figure 1 nanomaterials-13-02449-f001:**
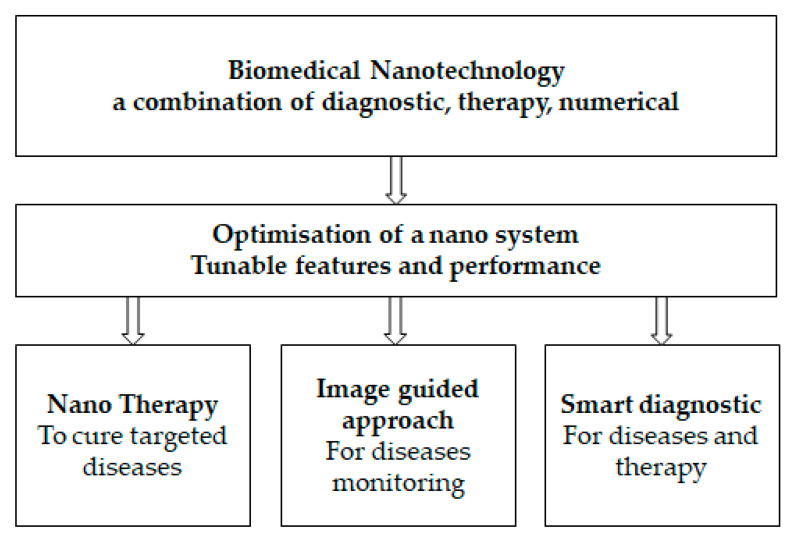
Biomedical nanotechnology diagram.

**Figure 2 nanomaterials-13-02449-f002:**
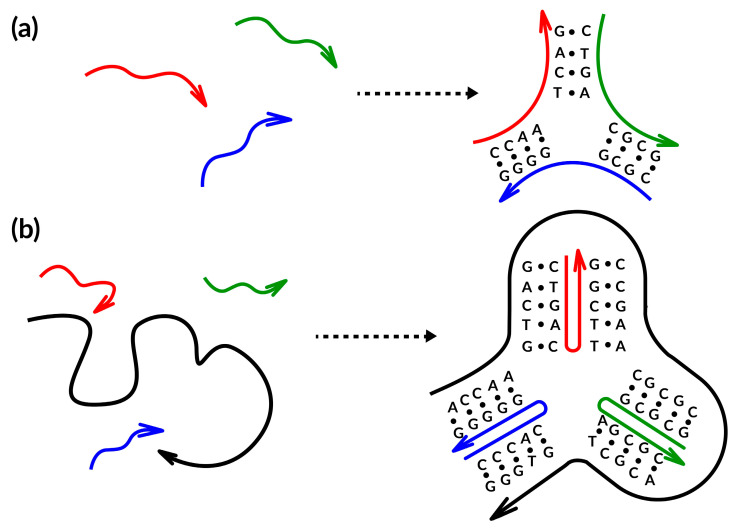
Schematic representation of the two primary design methods currently used to create DNA-NMs: (**a**) the tile-based; (**b**) the scaffold-based. The colored curved lines represent partially complementes oligos, while arrows point the oligos direction.

**Figure 3 nanomaterials-13-02449-f003:**
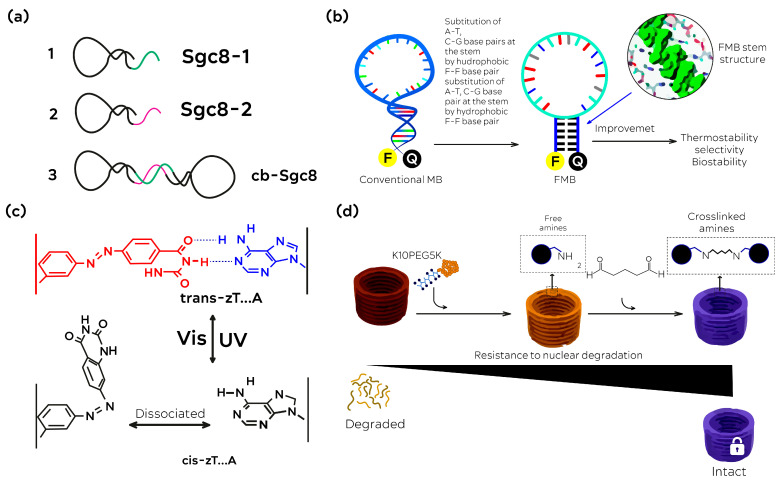
Improvement techniques for DNA-NMs and aptamers’ stability. (**a**) By ligating the 30 and 50 ends of the bivalent aptamers, a circle is created (1–3—steps of the process). (**b**) The use of fluorinated bases as useful nano molecules to enhance the stability of molecular beacons. (**c**) The new base pair pattern (zT-A) and the photosensitive base (zT). (**d**) PEGylated oligolysine-coated DNA-NMs with strong nuclease resistance are crosslinked with glutaraldehyde.

**Figure 4 nanomaterials-13-02449-f004:**
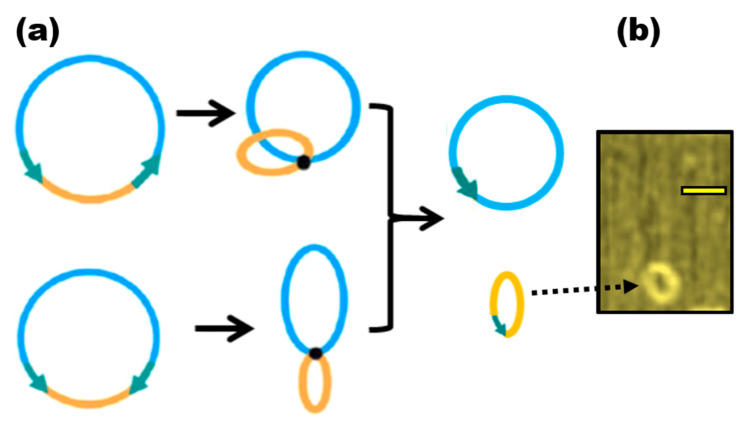
Microcircles generation with Cre recombinase. (**a**) The recombination either from two direct or inverted loxP sites (shown by jade arrows) results in excised microcircle. (**b**) AFM image of 146 bp microcircles from A.A.V.’s collection. The scale bar is 50 nm in length. The dotted arrow points to the microcircle.

**Figure 5 nanomaterials-13-02449-f005:**
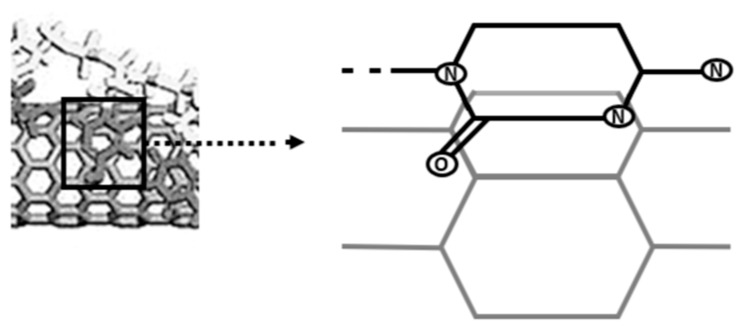
SWNTs and single-stranded DNA (ssDNA) have been combined in a theoretical model.

## Data Availability

Not applicable.

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
