# Peer review of "Recent Advances in DNA Nanomaterials"

_nanomaterials, 2023, doi:10.3390/nano13172449_

Round 1

Reviewer 1 Report

This manuscript by Bekkouche et al., entitled “Nanomanipulations with DNA”, purports to describe the application of nanomanipulations, techniques to manipulate 1-1000 nm DNA species, in fields ranging from medicine to energy. Think atomic force microscopy and optical tweezers. Rather, their abbreviated commentary provides but a cursory overview of several DNA-based nanomaterials – e.g., Seeman nucleic acid junctions, Wells triplet repeats, SELEX-derived aptamers, which have been well described in prior reviews. Nanomaterials, yes, but hardly nanomanipulation. This is compounded by the reuse of figures from several published articles (e.g., Fig. 1 & Ref. 15, Fig. 2 & Ref. 43, etc.]. Annotation and author modification is insufficient justification for their inclusion. In my understanding, a review should be a widely encompassing study of a titular concept, providing a wealth of information on prior art. They provide the background for current studies and when well done, foster ties between disparate concepts, leading to future insights. Such is not so for the present manuscript. Thus, I cannot recommend its publication in Nanomaterials.

Specific concerns are listed following.

[Title] “Nanomanipulations with DNA”. Misleading. Few DNA manipulations are described in this manuscript. “DNA Nanomaterials”, perhaps, but this has been covered by several prior reviews.

[Lines 33 - 36] The writing in this manuscript can be quite confusing at times. For example, “…switching between two states. or DNA [7-9] and is employed…” Note: numerous typos were found throughout this manuscript. They detracted from its content considerably.

[Section 2.1. Biomedical goals] While I am familiar with Ajeet Kaushik’s 2019 Frontiers in Nanotechnology viewpoint article, promising improved diagnostics and therapeutics provided by nanoscale entities, I do not agree with their direct ties to bioinformatics and the Internet of Medical Things. Needs far more elaboration to be satisfactory.

[Figure 1 and others] Most figures in this manuscript have been taken from published articles. Annotated, yes; author-manipulated, well. Still, they are often unnecessarily low-resolution and somewhat erroneous (“Smart diagnostics”). Better would be for the authors to create their own figures to illustrate the concept they are trying to communicate.

[Lines 92-94] (Nanoarchitectonics) …has the potential to revolutionize everything [33].” Such sweeping statements, filled with superlatives, have no place in an objective scientific review. This is not advertisements.

[Line 144] (Figure 2a) perhaps?

[Lines 173-175] What are you trying to say with these two opposing parenthetical statements? Confusing.

[Section 4.1.1 Plasmid microcircle excision] This was perhaps the most interesting section in this manuscript. Unfortunately, it was diminished by some very poor writing (e.g., lines 357-359). “(Non-viral vectors) …definition that occasionally eludes scientific humility.” Really? Also, what does the appended exclamatory phrase “sic!” mean?

[Figure 4b] As the only original data included in this manuscript, please provide a higher-resolution image.

[Conclusions] Part of this are hard to understand. “DNase extensively excretes in vivo”? Reword and provide references. Ligands on DNA nanostructures – how are these related to receptors on cell surfaces?

Hard to understand in many places (see aforementioned).

Author Response

2023-08-02

To Reviewer 1

Dear Reviewer:

Thank you so much for your efforts to improve the quality of our submission. As about your comments, let me address them according to their order in your review:

  1. Nanomaterials, yes, but hardly nanomanipulation.

We changed the title

  1. This is compounded by the reuse of figures from several published articles (e.g., Fig. 1 & Ref. 15, Fig. 2 & Ref. 43, etc.].

This is a review, so we modified figures, and hope that our input in newly made figures is enough to consider them sufficient.  

  1. [Title] “Nanomanipulations with DNA”. Misleading. Few DNA manipulations are described in this manuscript. “DNA Nanomaterials”, perhaps, but this has been covered by several prior reviews.

Changed accordingly.

  1. [Lines 33 - 36] The writing in this manuscript can be quite confusing at times. For example, “…switching between two states. or DNA [7-9] and is employed…” Note: numerous typos were found throughout this manuscript. They detracted from its content considerably.

Edited accordingly.

  1. [Section 2.1. Biomedical goals] While I am familiar with Ajeet Kaushik’s 2019 Frontiers in Nanotechnologyviewpoint article, promising improved diagnostics and therapeutics provided by nanoscale entities, I do not agree with their direct ties to bioinformatics and the Internet of Medical Things. Needs far more elaboration to be satisfactory.

Edited accordingly and we hope to satisfy reviewer’s point.

  1. [Figure 1 and others] Most figures in this manuscript have been taken from published articles. Annotated, yes; author-manipulated, well. Still, they are often unnecessarily low-resolution and somewhat erroneous (“Smart diagnostics”). Better would be for the authors to create their own figures to illustrate the concept they are trying to communicate.

This is a review, so we modified before  figures, and hope that our input in newly made figures is enough to consider them sufficient.   

[Lines 92-94] (Nanoarchitectonics) …has the potential to revolutionize everything [33].” Such sweeping statements, filled with superlatives, have no place in an objective scientific review. This is not advertisements.

Edited accordingly and we hope to satisfy reviewer’s point. 

[Line 144] (Figure 2a) perhaps?

Corrected

[Lines 173-175] What are you trying to say with these two opposing parenthetical statements? Confusing.

Corrected

[Section 4.1.1 Plasmid microcircle excision] This was perhaps the most interesting section in this manuscript. Unfortunately, it was diminished by some very poor writing (e.g., lines 357-359). “(Non-viral vectors) …definition that occasionally eludes scientific humility.” Really? Also, what does the appended exclamatory phrase “sic!” mean?

Corrected 

[Figure 4b] As the only original data included in this manuscript, please provide a higher-resolution image.

This is a real microcircle, and therefore it looks thicker and fuzzier, that AFM images of regular plasmid DNA which is 20 times longer (and looks therefore 20 times thinner).

[Conclusions] Part of this are hard to understand. “DNase extensively excretes in vivo”? Reword and provide references. Ligands on DNA nanostructures – how are these related to receptors on cell surfaces?

Corrected

Please let us know if we could do something else to improve the quality of the submission.

Sincerely

Dr. Alex Vetcher

Reviewer 2 Report

I. Bekkouche and team reviewed a research topic, “Nanomanipulations with DNA”. The review mainly focused on the recent progress of nanomanipulations of DNA with biomedical and nanotechnological goals. In addition, the authors also discussed different technologies of DNA manipulations and their applications in vitro and in vivo

The authors provide an appreciable overview worth publishing in the MDPI Nanomaterials. However, we recommend a detailed revision addressing the following issues carefully before considering a possible publication. This will help your article to reach broader audiences and readers.

1.     Few long sentences were observed throughout the manuscript. Please trim them or divide them into two sentences. For example, abstract lines 11 to 15.

2.     Some acronyms or abbreviations were not expanded. All readers cannot immediately follow such acronyms or abbreviations.

Line 173: POI à the proteins of interest (POI)

3.     Redundant phrases with acronyms were observed. Please address them. They will increase the word count but not the scientific information. Show only once after their appearance in the manuscript.

Line 176 and 190: single-stranded DNA (ssDNA)

4.     We recommend the authors to introduce the following nanomanipulations of DNA with the appropriate references: packaging of DNA into nanostructures, adsorption of DNA onto nanoparticles, and spherical nucleic acids.

5.     Packaging of DNA into nanostructures:

Self-assembly of plasmid DNA (pDNA) and block copolymers composed of hydrophilic and cationic segments form a polyplex micelle and is regarded as a promising delivery system for gene delivery. Such controlled packaging of DNA, a giant polyelectrolyte with a contour length of micrometers, into nanosized structures is well correlated with its biological performance.

A)    Selectively spooling of single pDNA into a nanosized toroidal structure by polyion complexation with poly(ethylene glycol) (PEG)-conjugated polycationic polymers in 600 mM NaCl solution showed superior gene transduction capabilities compared to rod-like structures formed by quantized folding at 0 mM NaCl.

B)    Increasing the chain length of the PLys segment (degree of polymerization from 20 to 70) in a PEG-block-poly(amino acid) copolymer with a PEG molecular weight of 21-kDa folded the rigid double-stranded (ds)DNA pDNA into a short rod length of polyplex micelle. Regulation of the rod length below 200 nm was found to be critical for efficient cellular uptake, including epithelial and endothelial cells (Biomaterials 2014;35(20):5359-5368). However, the intrinsic rigidity of the double-helix structure of dsDNA limits its packaging to a size below the critical size of 50 nm as the rigidity of the dsDNA with a persistent length of 50 nm (Biophys J 2006;91(10):3607-16). To counteract this,  single-stranded (ss)DNA was prepared upon heat dissociation of linear dsDNA duplexes. Such ssDNA was complexed with PEG-block-poly(l-lysine) (PEG-PLys), resulting in a compact and spherical polyplex micelle, which distinguished from the rod-shaped PM formed from dsDNA. This ssDNA-loaded PM elicited therapeutic protein expression in tumor nests of intractable pancreatic cancer.

C)    Rod-like packaging of rigid double-stranded pDNA into a core of polyplex micelle showed enhanced gene expression compared to globular packaging (Biomacromolecules 2017;18(1):36-43).

D)    Nanomanipulating of double-stranded pDNA into the core surrounded by a thermoresponsive middle hydrophobic palisade and outer hydrophilic shell exhibited improved tolerance against nuclease attacks and polyion exchanges compared to those without hydrophobic palisades, thereby promoting gene transfection.

6.     Adsorption of DNA onto nanoparticles:

(A)    Single-stranded DNA was adsorbed by citrate-capped gold nanoparticles (AuNPs), resulting in increased AuNP stability, which forms the basis of a number of biochemical and analytical applications (Langmuir 2012;28(8):3896-902).

(B)    Freezing accelerated the adsorption of thiolated DNA strands onto AuNPs (Langmuir 2019;35(19):6476-6482).

(C)    Fluorescently-labeled DNA oligonucleotides were adsorbed onto iron oxide nanoparticles via the backbone phosphate and quench fluorescence. Arsenate ions exchange adsorbed DNA to increase fluorescence, allowing the detection of arsenate down to 300 nM (Chem Commun (Camb) 2014;50(62):8568-70)

7.     Spherical nucleic acids (SNAs):

SNAs are three-dimensional nanostructures consisting of nucleic acids that are densely functionalized and oriented spherically around a nanoparticle core. SNA enters the cells more rapidly and in higher quantities without the use of transfection agents by engaging scavenger receptors that facilitate caveolin-mediated endocytosis than that of its analogous, one-dimensional strands of the same sequence. These SNA are used for miRNA profiling (J Am Chem Soc 2012;134(3):1376-91), detection of mRNA in living cells using SNA-based NanoFlare constructs (Anal Chem 2012;84(4):2062-6), and immunostimulation or immunoregulation by engaging TLRs (Proc Natl Acad Sci U S A 2019;116(21):10473-10481).

8.     A recent study utilized DNA-origami templates to obtain precise control of the virus capsid assembly’s size and shape through electrostatic interactions between anionic DNA origami and cationic charges of the viral capsid. The obtained viral capsid coatings can shield the encapsulated DNA origami from degradation (10.1038/s41565-023-01443-x).

Author Response

To Reviewer 2

Dear Reviewer:

First of all, we would like to thank you for your evaluation of our humble contribution.Then we would like to express sincere gratitude for your efforts to improve the quality of our submission. As about your comments, let me address them according to their order in your review: 

  1. Few long sentences were observed throughout the manuscript. Please trim them or divide them into two sentences. For example, abstract lines 11 to 15.

Abstract is edited

  1. Some acronyms or abbreviations were not expanded. All readers cannot immediately follow such acronyms or abbreviations.

Line 173: POI à the proteins of interest (POI)

Corrected

  1. Redundant phrases with acronyms were observed. Please address them. They will increase the word count but not the scientific information. Show only once after their appearance in the manuscript.

Line 176 and 190: single-stranded DNA (ssDNA)

Corrected

  1. We recommend the authors to introduce the following nanomanipulations of DNA with the appropriate references: packaging of DNA into nanostructures, adsorption of DNA onto nanoparticles, and spherical nucleic acids.

Changed accordingly

  1. Packaging of DNA into nanostructures:

Self-assembly of plasmid DNA (pDNA) and block copolymers composed of hydrophilic and cationic segments form a polyplex micelle and is regarded as a promising delivery system for gene delivery. Such controlled packaging of DNA, a giant polyelectrolyte with a contour length of micrometers, into nanosized structures is well correlated with its biological performance.

  1. A)    Selectively spooling of single pDNA into a nanosized toroidal structure by polyion complexation with poly(ethylene glycol) (PEG)-conjugated polycationic polymers in 600 mM NaCl solution showed superior gene transduction capabilities compared to rod-like structures formed by quantized folding at 0 mM NaCl.
  2. B)    Increasing the chain length of the PLys segment (degree of polymerization from 20 to 70) in a PEG-block-poly(amino acid) copolymer with a PEG molecular weight of 21-kDa folded the rigid double-stranded (ds)DNA pDNA into a short rod length of polyplex micelle. Regulation of the rod length below 200 nm was found to be critical for efficient cellular uptake, including epithelial and endothelial cells (Biomaterials 2014;35(20):5359-5368). However, the intrinsic rigidity of the double-helix structure of dsDNA limits its packaging to a size below the critical size of ∼50 nm as the rigidity of the dsDNA with a persistent length of 50 nm (Biophys J 2006;91(10):3607-16). To counteract this,  single-stranded (ss)DNA was prepared upon heat dissociation of linear dsDNA duplexes. Such ssDNA was complexed with PEG-block-poly(l-lysine) (PEG-PLys), resulting in a compact and spherical polyplex micelle, which distinguished from the rod-shaped PM formed from ds This ssDNA-loaded PM elicited therapeutic protein expression in tumor nests of intractable pancreatic cancer.
  3. C)    Rod-like packaging of rigid double-stranded pDNA into a core of polyplex micelle showed enhanced gene expression compared to globular packaging (Biomacromolecules 2017;18(1):36-43).
  4. D)    Nanomanipulating of double-stranded pDNA into the core surrounded by a thermoresponsive middle hydrophobic palisade and outer hydrophilic shell exhibited improved tolerance against nuclease attacks and polyion exchanges compared to those without hydrophobic palisades, thereby promoting gene transfection.

All recommended publications were included with the description

  1. Adsorption of DNA onto nanoparticles:

(A)    Single-stranded DNA was adsorbed by citrate-capped gold nanoparticles (AuNPs), resulting in increased AuNP stability, which forms the basis of a number of biochemical and analytical applications (Langmuir 2012;28(8):3896-902).

(B)    Freezing accelerated the adsorption of thiolated DNA strands onto AuNPs (Langmuir 2019;35(19):6476-6482).

(C)    Fluorescently-labeled DNA oligonucleotides were adsorbed onto iron oxide nanoparticles via the backbone phosphate and quench fluorescence. Arsenate ions exchange adsorbed DNA to increase fluorescence, allowing the detection of arsenate down to 300 nM (Chem Commun (Camb) 2014;50(62):8568-70)

All recommended publications were included with the description

  1. Spherical nucleic acids (SNAs):

SNAs are three-dimensional nanostructures consisting of nucleic acids that are densely functionalized and oriented spherically around a nanoparticle core. SNA enters the cells more rapidly and in higher quantities without the use of transfection agents by engaging scavenger receptors that facilitate caveolin-mediated endocytosis than that of its analogous, one-dimensional strands of the same sequence. These SNA are used for miRNA profiling (J Am Chem Soc 2012;134(3):1376-91), detection of mRNA in living cells using SNA-based NanoFlare constructs (Anal Chem 2012;84(4):2062-6), and immunostimulation or immunoregulation by engaging TLRs (Proc Natl Acad Sci U S A 2019;116(21):10473-10481).

Corrected accordingly

  1. A recent study utilized DNA-origami templates to obtain precise control of the virus capsid assembly’s size and shape through electrostatic interactions between anionic DNA origami and cationic charges of the viral capsid. The obtained viral capsid coatings can shield the encapsulated DNA origami from degradation (10.1038/s41565-023-01443-x).

Edited accordingly

Please let us know if we can do something else to improve the submission quality

Sincerely

Dr. Alex Vetcher

Round 2

Reviewer 1 Report

This is a revised manuscript by Bekkouche et al. The authors have addressed some of the prior concerns. However, it is this reviewer’s opinion that it still may not meet the standards for publication in Nanomaterials.

Specific concerns are listed following.

[Title] Revised title, “Recent Advances in DNA Nanomaterials” is appropriate, given the content of the review.

[Abstract] Abstract is highlighted, indicative of changes. These were not apparent. It is still focused on nanomanipulations, which is not really reflected in the text.

[Text] Much of the studies on DNA nanomaterials are referred to in the text as DNA nanomanipulations (NM). But they are not. Better would be to refer to them as DNA nanomaterials (NM) throughout.

[Lines 33 - 36] Several typographical errors have been corrected in the text. Unfortunately, not all (e.g., Line 399 sic!). This manuscript requires careful proofreading before publication.

[Section 2.1. Biomedical goals] Requested elaboration on the direct ties between DNA nanomaterials and bioinformatics and/or the Internet of Medical Things was not provided. This is troublesome, as often the authors state with a few words (“Edited accordingly”) that adequate changes have been made when in fact, they have not. Needs far more elaboration to be satisfactory.

[Figure 1 and others] As noted previously, most figures in this manuscript have been taken directly from published articles. Perhaps that can be tolerated in a review, so long as the copyrights have been secured. However, their low resolution is unacceptable. Better would be for the authors to create their own figures to illustrate the concepts they are trying to communicate.

[Lines 92-94] More appropriate, but grammatically incorrect (run-on sentence).

[Section 3] NM?

[Lines 165-175] No longer as confusing but, no references?

[Sections 3.4 & 3.5] Welcome additions.

[Sections 4.3 & 4.4] Welcome additions.

Manuscript would benefit from careful proofreading.

Author Response

To Reviewer 1

Dear Reviewer:

Let me express our sincere gratitude for your efforts to improve the quality of our submission. As about your comments, let me respond on them in the order in your review:

  1. [Title] Revised title, “Recent Advances in DNA Nanomaterials” is appropriate, given the content of the review.

 We changed the Title accordingly

  1. [Abstract] Abstract is highlighted, indicative of changes. These were not apparent. It is still focused on nanomanipulations, which is not really reflected in the text.

 We changed the Abstract accordingly

  1. [Text] Much of the studies on DNA nanomaterials are referred to in the text as DNA nanomanipulations (NM). But they are not. Better would be to refer to them as DNA nanomaterials (NM) throughout.

We incorporated corrections accordingly

  1. [Lines 33 - 36] Several typographical errors have been corrected in the text. Unfortunately, not all (e.g., Line 399 sic!). This manuscript requires careful proofreading before publication.

 We corrected all typos, which we were able to find

  1. [Section 2.1. Biomedical goals] Requested elaboration on the direct ties between DNA nanomaterials and bioinformatics and/or the Internet of Medical Things was not provided. This is troublesome, as often the authors state with a few words (“Edited accordingly”) that adequate changes have been made when in fact, they have not. Needs far more elaboration to be satisfactory.

I hope that currently it looks more highlighted

  1. [Figure 1 and others] As noted previously, most figures in this manuscript have been taken directly from published articles. Perhaps that can be tolerated in a review, so long as the copyrights have been secured. However, their low resolution is unacceptable. Better would be for the authors to create their own figures to illustrate the concepts they are trying to communicate.

Now the figures contain much more originality and they are of much higher quality

  1. [Lines 92-94] More appropriate, but grammatically incorrect (run-on sentence).

Corrected

  1. [Section 3] NM?

Corrected

  1. [Lines 165-175] No longer as confusing but, no references?

 Corrected

  1. [Sections 3.4 & 3.5] Welcome additions.

 Corrected

  1. [Sections 4.3 & 4.4] Welcome additions.

 Corrected

Please, let us know if we can do something else to improve the quality of the submission.

Sincerely

Dr. Alex Vetcher

Reviewer 2 Report

The authors did not clearly understand the suggestions and comments given in the previous review. 

The scientific information is misleading. Please thoroughly understand the articles and discuss them in the review. Please read the suggestions and comments given by the reviewer again and address them all. 

Does single-stranded DNA form toroids?

A single molecule of double-stranded pDNA is packaged into a toroidal structure under high NaCl concentration upon complexation with PEG-block catiomers. 

Double-stranded, supercoiled plasmid DNA will be crumpled and form globular-like structures upon complexation with PEG-block-polycationic polymers when the reduced tethering density of PEG is below 1. But the authors mentioned that ssDNA forms a rod-like structure.

dsDNA also forms a rod-like structure when the reduced tethering density of PEG is above 1.

Heat melting of linear dsDNA separates the strands into ssDNA. Immediate complexation with PEG-block-polycationic polymers packages the ssDNA into sphere-like structures. 

Rod-like packaging of ds supercoiled pDNA using cyclic-RGD ligand conjugated PEG-block-poly(l-lysine-thiol) showed significant antitumor activity against pancreatic adenocarcinoma, one of intractable cancer. Highlighting the biomedical significance of such DNA nanoparticles will be interesting to the readers. PLEASE ADDRESS ALL the suggestions and comments. 

Author Response

To Reviewer 2

Dear Reviewer:

Let me express our sincere gratitude for your efforts to improve the quality of our submission. As about your comments, let me respond on them in the order in your review:

The authors did not clearly understand the suggestions and comments given in the previous review. The scientific information is misleading. Please thoroughly understand the articles and discuss them in the review. Please read the suggestions and comments given by the reviewer again and address them all. 

We re-edited the submission.

  1. Few long sentences were observed throughout the manuscript. Please trim them or divide them into two sentences. For example, abstract lines 11 to 15.

The submission was re-edited accordingly

  1. Some acronyms or abbreviations were not expanded. All readers cannot immediately follow such acronyms or abbreviations.

Line 173: POI à the proteins of interest (POI)

Corrected

  1. Redundant phrases with acronyms were observed. Please address them. They will increase the word count but not the scientific information. Show only once after their appearance in the manuscript.

Line 176 and 190: single-stranded DNA (ssDNA)

Corrected

  1. We recommend the authors to introduce the following nanomanipulations of DNA with the appropriate references: packaging of DNA into nanostructures, adsorption of DNA onto nanoparticles, and spherical nucleic acids.

Changed accordingly

  1. Packaging of DNA into nanostructures:

Self-assembly of plasmid DNA (pDNA) and block copolymers composed of hydrophilic and cationic segments form a polyplex micelle and is regarded as a promising delivery system for gene delivery. Such controlled packaging of DNA, a giant polyelectrolyte with a contour length of micrometers, into nanosized structures is well correlated with its biological performance.

  1. A) Selectively spooling of single pDNA into a nanosized toroidal structure by polyion complexation with poly(ethylene glycol) (PEG)-conjugated polycationic polymers in 600 mM NaCl solution showed superior gene transduction capabilities compared to rod-like structures formed by quantized folding at 0 mM NaCl.
  2. B) Increasing the chain length of the PLys segment (degree of polymerization from 20 to 70) in a PEG-block-poly(amino acid) copolymer with a PEG molecular weight of 21-kDa folded the rigid double-stranded (ds)DNA pDNA into a short rod length of polyplex micelle. Regulation of the rod length below 200 nm was found to be critical for efficient cellular uptake, including epithelial and endothelial cells (Biomaterials 2014;35(20):5359-5368). However, the intrinsic rigidity of the double-helix structure of dsDNA limits its packaging to a size below the critical size of ∼50 nm as the rigidity of the dsDNA with a persistent length of 50 nm (Biophys J 2006;91(10):3607-16). To counteract this, single-stranded (ss)DNA was prepared upon heat dissociation of linear dsDNA duplexes. Such ssDNA was complexed with PEG-block-poly(l-lysine) (PEG-PLys), resulting in a compact and spherical polyplex micelle, which distinguished from the rod-shaped PM formed from ds This ssDNA-loaded PM elicited therapeutic protein expression in tumor nests of intractable pancreatic cancer.
  3. C) Rod-like packaging of rigid double-stranded pDNA into a core of polyplex micelle showed enhanced gene expression compared to globular packaging (Biomacromolecules 2017;18(1):36-43).
  4. D) Nanomanipulating of double-stranded pDNA into the core surrounded by a thermoresponsive middle hydrophobic palisade and outer hydrophilic shell exhibited improved tolerance against nuclease attacks and polyion exchanges compared to those without hydrophobic palisades, thereby promoting gene transfection.

All recommended publications were included with the description

  1. Adsorption of DNA onto nanoparticles:

(A) Single-stranded DNA was adsorbed by citrate-capped gold nanoparticles (AuNPs), resulting in increased AuNP stability, which forms the basis of a number of biochemical and analytical applications (Langmuir 2012;28(8):3896-902).

(B) Freezing accelerated the adsorption of thiolated DNA strands onto AuNPs (Langmuir 2019;35(19):6476-6482).

(C) Fluorescently-labeled DNA oligonucleotides were adsorbed onto iron oxide nanoparticles via the backbone phosphate and quench fluorescence. Arsenate ions exchange adsorbed DNA to increase fluorescence, allowing the detection of arsenate down to 300 nM (Chem Commun (Camb) 2014;50(62):8568-70)

All recommended publications were included with the description

  1. Spherical nucleic acids (SNAs):

SNAs are three-dimensional nanostructures consisting of nucleic acids that are densely functionalized and oriented spherically around a nanoparticle core. SNA enters the cells more rapidly and in higher quantities without the use of transfection agents by engaging scavenger receptors that facilitate caveolin-mediated endocytosis than that of its analogous, one-dimensional strands of the same sequence. These SNA are used for miRNA profiling (J Am Chem Soc 2012;134(3):1376-91), detection of mRNA in living cells using SNA-based NanoFlare constructs (Anal Chem 2012;84(4):2062-6), and immunostimulation or immunoregulation by engaging TLRs (Proc Natl Acad Sci U S A 2019;116(21):10473-10481).

Corrected accordingly

  1. A recent study utilized DNA-origami templates to obtain precise control of the virus capsid assembly’s size and shape through electrostatic interactions between anionic DNA origami and cationic charges of the viral capsid. The obtained viral capsid coatings can shield the encapsulated DNA origami from degradation (10.1038/s41565-023-01443-x).

Edited accordingly

Does single-stranded DNA form toroids?

A single molecule of double-stranded pDNA is packaged into a toroidal structure under high NaCl concentration upon complexation with PEG-block catiomers. 

Double-stranded, supercoiled plasmid DNA will be crumpled and form globular-like structures upon complexation with PEG-block-polycationic polymers when the reduced tethering density of PEG is below 1. But the authors mentioned that ssDNA forms a rod-like structure.

dsDNA also forms a rod-like structure when the reduced tethering density of PEG is above 1.

Heat melting of linear dsDNA separates the strands into ssDNA. Immediate complexation with PEG-block-polycationic polymers packages the ssDNA into sphere-like structures. 

Rod-like packaging of ds supercoiled pDNA using cyclic-RGD ligand conjugated PEG-block-poly(l-lysine-thiol) showed significant antitumor activity against pancreatic adenocarcinoma, one of intractable cancer. Highlighting the biomedical significance of such DNA nanoparticles will be interesting to the readers. PLEASE ADDRESS ALL the suggestions and comments. 

We expanded related sections accordingly.

Please let us know if we can do something else to improve the quality of our submission

Regards

Dr. Alex Vetcher

Round 3

Reviewer 1 Report

The authors have adequately addressed all my concerns. I am especially pleased with the improved figures. In my opinion, this manuscript is now suitable for publication in Nanomaterials.

Generally good.

Author Response

Dear Reviewer:

Thank you so much for your high evaluation of our humble contribution.

Sincerely

Dr. Alex Vetcher

Reviewer 2 Report

The correct citation of the pDNA toroidal structure is reference [127]. However, in the current manuscript, the authors cited reference [128]. We mentioned the same in the previous revision. The authors again failed to cite the correct reference. 

pDNA self-assembly by PEG-polycationic block copolymers results in either toroidal structures, rod structures, or globular structures depending on salt concentration and strandedness of DNA. For example, the complexation of double-stranded pDNA with PEG-polycation in 600 mM NaCl forms toroidal structures. Without salt, the same double-stranded DNA form rod-shaped structures. Single-stranded DNA forms spherical structures. Cite the information with correct references. 

Although the authors mentioned that they discussed all the scientific information suggested by the reviewer, they did not discuss the above information. Please revisit the suggestions and comments made by the reviewer carefully.

Author Response

Dear Reviewer:

Thank you so much for your time and efforts. Let me address your comments in their order in your review:

  1. The correct citation of the pDNA toroidal structure is reference [127]. However, in the current manuscript, the authors cited reference [128]. We mentioned the same in the previous revision. The authors again failed to cite the correct reference.

Corrected

2. pDNA self-assembly by PEG-polycationic block copolymers results in either toroidal structures, rod structures, or globular structures depending on salt concentration and strandedness of DNA. For example, the complexation of double-stranded pDNA with PEG-polycation in 600 mM NaCl forms toroidal structures. Without salt, the same double-stranded DNA form rod-shaped structures. Single-stranded DNA forms spherical structures. Cite the information with correct references. 

Corrected

3. Although the authors mentioned that they discussed all the scientific information suggested by the reviewer, they did not discuss the above information. Please revisit the suggestions and comments made by the reviewer carefully.

I hope that the current version after edition is satisfactory 

Thank you again 

Sincerely

Dr. Alex Vetcher

Round 4

Reviewer 2 Report

 Accepted in present form